# Recognizing and Appreciating the Partner’s Support Protects Relationship Satisfaction during Cardiac Illness

**DOI:** 10.3390/jcm13041180

**Published:** 2024-02-19

**Authors:** Giada Rapelli, Silvia Donato, Emanuele Maria Giusti, Giada Pietrabissa, Miriam Parise, Ariela Francesca Pagani, Chiara A. M. Spatola, Anna Bertoni, Gianluca Castelnuovo

**Affiliations:** 1Department of Medicine and Surgery, University of Parma, 43125 Parma, Italy; giada.rapelli@unipr.it; 2Department of Psychology, Catholic University of Sacred Heart, 20123 Milan, Italy; silvia.donato@unicatt.it (S.D.); anna.bertoni@unicatt.it (A.B.); gianluca.castelnuovo@auxologico.it (G.C.); 3Family Studies and Research University Centre, Catholic University of Sacred Heart, 20123 Milan, Italy; 4EPIMED Research Center, Department of Medicine and Surgery, University of Insubria, 21100 Varese, Italy; 5Department of Humanities, University of Urbino Carlo Bo, 61029 Urbino, Italy; 6Department of Clinical and Experimental Medicine, University of Messina, 98122 Messina, Italy; chiara.spatola@unime.it

**Keywords:** cardiac illness, dyadic coping, couple, stress, psychological distress, relationship satisfaction

## Abstract

Background: This study aimed to examine whether dyadic coping (DC) is associated with relationship satisfaction (RS) among couples facing cardiac diseases. Furthermore, the moderating role of both partners’ anxiety and depression was tested. Methods: One hundred cardiac patients (81.5% men) and their partners (81.5% women) completed a self-report questionnaire during hospitalization. The Actor–Partner Interdependence Model (APIM) and moderation analyses were used to assess the above associations. Results: Results showed that positive DC was significantly related to higher levels of RS, and negative DC was related to lower levels of RS. Furthermore, patient and partner psychological distress significantly moderated the link between DC and RS: patient-perceived positive DC was associated with higher partner RS when partner depression was high; partner-perceived positive DC was associated with higher patient RS when patient anxiety was low; patient-perceived negative DC has associated with lower patient RS when patient anxiety and depression were high. Conclusion: This study showed that positive DC is associated with a more satisfying relationship and identified under what conditions of cardiac-related distress this can happen. Furthermore, this study underlined the importance of examining DC in addition to the individual coping skills as a process pertaining to personal well-being and couple’s outcomes.

## 1. Introduction

The diagnosis of a cardiac illness is a stressful condition that negatively influences the patient’s life [1]. It demands many lifestyle changes (e.g., a diet, physical activity, smoking and alcohol consumption, medical check-ups, and prescription drug compliance). As a result, cardiac patients could have higher rates of mood disorders such as depression [1,2] and anxiety [3] than healthy individuals. These disorders are risk factors for poor prognosis and adherence, high mortality, and low quality of life [4,5,6,7].

Furthermore, cardiac-related distress impacts not only the patient but also his/her spouse and their relationship. On the one hand, partners might show intense psychological distress due to caregiving activities also affecting the balance between given and received support in the couple [8,9,10]. On the other hand, supportive partner behaviors could have a positive effect on patient psychological reaction [11,12], and on cardiac self-management outcomes, for example, by increasing the levels of medication adherence [13] and patient engagement [13,14]. These dynamics of interdependence and mutual influence between partners could be explained by the concept of dyadic coping (DC; [15,16]), i.e., the couple’s level of ability to cope with daily or major stress situations together. DC is an interpersonal process composed of: (a) stress expression by the stressed person, (b) the perception of stress by the partner, and (c) the partner’s coping reactions [15,16]. The role of this process in daily as well as major stressor events and its effect on partners’ health and couple variables have been largely investigated in the general population (e.g., [17,18,19]) and in chronic patients [20,21,22], but only partially explored in cardiac patients and their partners (e.g., [9,12,13]).

According to the Systemic Transactional Model (STM) of Bodenmann [23], a distinction is made between positive and negative DC. Positive DC can be implemented either through emotionally oriented strategies, i.e., aimed at reducing the partner‘s state of emotional distress, or through problem-oriented strategies, which aim at solving the problem that gave rise to stress. It includes supportive and delegated dyadic coping: supportive dyadic coping, in particular, occurs when a partner expresses understanding and solidarity with the other, or when the partner provides information and practical suggestions to the other. In delegated dyadic coping, a partner explicitly asks for the other‘s help to relieve and alleviate the stress. Negative DC includes ambivalent dyadic coping, superficial dyadic coping, and hostile dyadic coping. All these forms of negative dyadic coping can be considered coping responses somehow aimed at supporting the partner but implemented inefficiently or incompetently. Ambivalent dyadic coping occurs when the partner provides support expressing reluctance to help, because, for example, the requested support is seen as unnecessary and/or the other partner is seen as inferior or incompetent. In this case, one partner offers his/her help but has the feeling that the other should not have asked for it. Superficial dyadic coping, on the other hand, occurs when the partner provides support to the other but does so in an insincere and untrue way, for example by asking what the problem is, but without listening to the other‘s response or embracing him or her without being emotionally involved. Hostile dyadic coping is when one partner reacts to the communication of the other‘s stress with negative comments or behavior, for example, by not taking the problem seriously or by distancing him/herself.

According to the literature, positive DC was found to be associated with better individual outcomes (e.g., positive emotions, better psychological well-being, more physical activity, and higher life satisfaction); conversely, negative DC was found to be harmful in the general population and in chronic patients [24] and these trends also emerged among cardiac patients [12,13]. Moreover, DC was shown to be highly predictive for relationship functioning, and its stability; in particular, positive DC is associated with better marital adjustment, tenderness and togetherness, higher sexual satisfaction, and long-lasting marriages [25,26,27,28].

The STM of Bodenmann [23] supported the strong associations between DC and relationship functioning (e.g., marital satisfaction and marital communication). Furthermore, it seems that DC is a stronger predictor of relationship quality than individual coping strategies [28,29]. In particular, in a meta-analysis, Falconier et al. [30] documented that all positive and common DC strategies were significant positive predictors of Relationship Satisfaction (RS), whereas all negative DC strategies were significant negative predictors. Partners’ satisfaction with their relationship was largely associated with their perception of the couple’s DC, regardless of partners’ gender, age, educational level, and couple relationship duration [24,31]. Exploring the RS is important because it is a key component of life satisfaction [32], strongly associated with individual psychological and physical well-being [33,34]. Furthermore, relationship quality and stability are considered protective factors both for the cardiac patient in terms of survival [35,36] and mental health [37] and also for the caregiver‘s psychological well-being [37,38]. In fact, in the cardiac disease context, it is important to consider not only the caregiver as a provider of cure and the patient as a receiver but also their specific contribution to the other and to the relationship more generally because, despite the presence of a cardiac illness, the patient and the caregiver continue to be partners with relational needs, for example, closeness and intimacy [8,39,40].

The link between DC and RS has been widely investigated in different types of couples (e.g., late adolescent couples, newly married couples, couples during pregnancy, and older couples) [18,41,42,43,44] and considering different critical events and/or transitions that the couple may face, such as an illness [20,21,45,46,47,48]. For the reasons above, it is important to confirm the role of DC as a strong predictor of RS also among cardiac patients and their spouses, as well as explore their reciprocal influences using a dyadic lens.

Moreover, although various studies have reported strong associations between DC and RS, none have considered the moderating effects of distress among cardiac patients and their partners. Indeed, cardiac disease is a stressful situation for both patient [5,49] and partner [8,50], and their level of individual distress can likely interfere with their ability to cope effectively as a couple. Levels of the patient and partner’s distress in terms of depression and anxiety may influence how the cardiac illness is managed [12] and some authors also demonstrated the strong association between individual distress and relational well-being of partners in non-clinical samples [33,51], as well as among couples coping with chronic illness [13,38,47]. Stress can exert its negative impact on the couple in at least four different ways: by decreasing the time partners spend together, which in turn weakens partners’ feelings of mutuality; by decreasing self-disclosure and effective communication; by increasing the likelihood that partners’ problematic personality traits (e.g., anxiety, dominance, rigidity, and stubbornness) will be expressed and impact the other; by increasing the risk of deleterious health outcomes, such as mood or sleep disorders [52]. As such, stressful experiences may negatively affect not only the partners’ individual well-being, but also erode their RS over time, and eventually lead to divorce [52,53,54].

Given the lack of studies assessing the moderating effect of psychological distress among partners dealing with cardiac illness, in this study, the relationship between patient and partner DC with their RS was tested and the moderating role of both partner’s anxiety and depression was taken into account. As argued by Randall and Bodenmann [54], it is important to consider the moderating effect of partners’ distress affecting the couple’s functioning because, during an illness, both partners have to manage the distress, make lifestyle changes, and cope with the burdens and disruptions the disease imposes [8,9,55].

Based on previous research, the following hypotheses were tested:

**Hypothesis** **1.**
*Positive DC is expected to be related to higher levels of RS at intrapersonal and interpersonal levels [18,30,42]; conversely, negative DC is assumed to be related to lower levels of RS at intrapersonal and interpersonal levels [30].*


**Hypothesis** **2.**
*The association between positive DC and RS is predicted to be stronger for patients and partners who reported low psychological distress and weaker if patient and partner psychological distress are high. The opposite is expected for negative DC.*


Figure 1 depicts the theoretical model tested in the following analyses.

## 2. Materials and Methods

### Participant and Procedure

One hundred mixed-sex couples participated in this study. The couples were enrolled in six different hospitals in Northern Italy (Ospedale Maggiore di Lodi (LO), Ospedale Bolognini di Seriate (BG), Ospedale di Chiari (BS), Ospedale Maggiore della Carità di Novara (NO), Istituto Auxologico Italiano—Ospedale San Luca (MI), and Istituto Auxologico Italiano—Ospedale San Giuseppe di Piancavallo (VB)) two days before patient discharge from hospitalization for an acute cardiac event (on average, the hospitalization was for 14 days across the centers). Criteria for study inclusion were as follows: (1) having a partner; (2) admission for acute cardiovascular disease (e.g., ischemic heart diseases, like myocardial infarction and acute coronary syndrome); (3) no mental disability, assessed with a short version of the Mini-Mental State Examination (MMSE; the scores range from 0 to 25 and the cut-off score used for the present study was 20); (4) ability to understand the Italian language. Upon enrollment, couples were asked to independently fill out a set of two questionnaires, one for the patient and one for the partner. Signed informed consent was obtained from all participants. The Psychology Research Ethics Committee of the Catholic University of Sacred Heart of Milan approved this study (Protocol number 37-18).

## 3. Measures

Both patients’ and partners’ questionnaires included items assessing socio-demographic variables (i.e., age, sex, marital status, and relationship duration, education level, and employment status) and only patient clinical data (e.g., blood pressure, comorbidities BMI). The questionnaires also included the following measurement instruments.

### 3.1. Dyadic Coping

To assess DC, the Italian version of the DC questionnaire was used [23,56]. This 41-item questionnaire measures the propensity of each partner to offer help, emotional support, and empathy in response to the other’s expression of stress. The scale considers different forms of DC. For the present study, we focused on both positive (e.g., “My partner is on my side and tells me that he/she knows how it feels to be stressed and that he/she cares about me”) and negative (e.g., “My partner helps me, but does so unwillingly and unmotivated”) partner DC responses (patient- and partner-perceived DC). The items are administered on a 5-point scale (from 1 = never to 5 = very often). Internal consistency (Cronbach’s α) in our sample ranged between α = 0.65 and α = 0.88 for the patient subscales and between α = 0.65 and α = 0.89 for the partner subscales.

### 3.2. Relationship Satisfaction

RS was measured by the Quality of Marriage Index [57]. Five items are administered on a 7-point scale (from 1 = totally disagree to 7 = totally agree). An item example is “The relationship with my partner makes me happy”. By averaging the five items, a global index of patient and partner RS was created, with higher scores referring to higher satisfaction. Cronbach’s alpha for the current study was 0.93 for both patients and partners.

### 3.3. Psychological Distress

Patient and partner psychological distress were measured by the 25-item version of the Hopkins Symptom Checklist (HSCL-25; [58]). The scale consists of 25 items measuring symptoms of anxiety (e.g., In the past week, to what extent did you worry or stress for this symptom “Difficulty in falling asleep and in sleeping?”.) and depression (e.g., In the past week, to what extent did you worry or stress for this symptom “Feeling no interests in things”). Participants were asked to rate symptoms experienced during the past week as ranging from 1 = never to 4 = often. The total score of the scale was computed by averaging the items for the two subscales: a high score implied higher psychological distress. The cut-off clinical score was set at 1.70, according to the validation study [58]. The HSCL-25 correlated highly with the standard 58-item version [59] and showed good predictive and discriminative validity [58]. This scale has not yet been used in the cardiac population [60]. Cronbach’s alpha for the anxiety subscale in the current study was 0.80; and for the depression, the subscale was 0.83.

## 4. Analytical Strategy

The descriptive statistics were performed separately for patients and partners. Categorical variables were described using percentages, continuous variables using means, and standard deviations. Paired-sample *t*-tests were used to analyze differences between patients and partners regarding positive and negative DC, RS, anxiety, and depression. Pearson’s correlations were used to test the associations between the study variables.

To address data interdependence and treat the dyad as the unit of analysis, the Actor–Partner Interdependence Model (APIM; [61]) was used within a path analysis approach to model the associations between the study variables. The APIM is a dyadic data analytic approach that enables estimating effects for both members of the couple simultaneously while controlling for the interdependence between them [61]. In the APIM, each model includes the predictor and outcome variables measured in both couple members. Actor and partner effects are simultaneously estimated. The actor effect is the intrapersonal relationship between the predictor and the outcome and is represented for each member of the couple by a path linking the predictor with the outcome measured in the same member. The partner effect is the interpersonal relationship between the predictor and the outcome and is represented for each member of the couple by a path linking the predictor measured in that member and the outcome measured in the other member. Both the predictors and the error terms of the outcomes were allowed to be correlated.

Preliminarily, to test whether the dyad members were empirically distinguishable, we followed the strategy proposed by Kenny et al. [61] and ran a path analysis model with variances and covariances in the variables of interest constrained to be equal for patients and partners. The unconstrained model presented a significantly better fit than the constrained one (*x*2 = 30.323, *p* < 0.001), which means that dyads could be considered empirically distinguishable.

Variables were included in the APIM based on the theoretical model represented in Figure 1. To test the first hypothesis, we tested the actor and partner effects representing the relationships between patients’ and partners’ positive DC and RS. This model was then repeated using negative DC as the independent variable.

To test the second hypothesis, we used the Actor–Partner Interdependence Moderation Model (APIMoM; [62]) to test the moderation of patient and partner distress only for significant effects of the APIM. Briefly, the APIMoM is an extension of the APIM which includes the effect of the interaction terms between the predictor and one or more moderators. Regarding the assumptions, it is assumed that there is no measurement error in X and M. Additionally, it is assumed that there are no unmeasured common causes (i.e., confounders) between X and M, between X and Y, and between M and Y. It is assumed that Y does not cause X or M and that M does not cause X. Finally, it is assumed that the relationships between X and Y and between M and Y are linear and that the interaction between X and Y is also linear. In this study, we tested eight separate APIMoMs. In all these models, RS was the outcome, and the moderating effects were calculated for all actor and partner effects. In model 1, the independent variable was positive DC and the moderator was patient depression. In model 2, the independent variable was positive DC and the moderator was partner depression. Models 3 and 4 were similar to models 1 and 2, respectively, with the exception that the independent variable was negative DC. Models 5, 6, 7, and 8 were similar to models 1, 2, 3, and 4, respectively, with the exception that anxiety was the moderator variable. Moderation was identified in case the interaction coefficients were significant.

In case of significant moderation effects, simple slope analyses were performed. Since the moderator was a continuous variable, the coefficients of the independent variables at −1 and +1 standard deviation from the mean of the moderator were extracted.

Before being included in the APIM and APIMoM, all metric variables were z-standardized using the means and standard deviations of the whole sample of patients and partners [61]. Therefore, the coefficients of the models should be interpreted as standardized coefficients relative to the grand mean. No confounder was included in the models. The significance threshold was set at *p* = 0.05.

The descriptive statistics, the *t*-tests, and Pearson’s correlations were computed using the software IBM SPSS version 22.0 (SPSS Inc. Chicago IL, USA). The APIM and the APIMoM analyses were performed using AMOS 19.0 [63].

## 5. Results

### 5.1. Preliminary Data Analyses

Table 1 shows the characteristics of the couples included in this study. Most patients are men, and patients and partners are on average 60 years old. On average, their relationship duration is 36.34 years. All the participants are White.

Table 2 shows the means, standard deviations, and paired-sample *t*-tests between patients’ and partners’ variables. The positive DC (PDC) perceived by the patient and by the partner is high for both patients and partners as compared to the scale range. Conversely, the negative DC (NDC) perceived by the patient and by the partner is low as compared to the scale range. The RS is high for both patients and partners as compared to the scale range. Concerning psychological distress, levels of anxiety and depression are high and just below the cut-off score of the scale for clinical significance, except for the partner’s anxiety exceeding the clinical cut-off score.

The only significant paired-samples *t*-test shows that patient-perceived positive DC is significantly higher than the partner-perceived one [*t* (98) = 4.21, *p* < 0.001]. No other significant difference is found.

Table 3 shows the correlations between the study variables. Patient and partner-perceived PDC show a strong and positive correlation with RS. Patient and partner-perceived NDC show a moderate and negative correlation with RS. There is a moderate and positive correlation between patient and partner RS. The patient and partner psychological distress are strongly correlated with each other and with RS.

### 5.2. Relationship between Positive and Negative DC with RS

The results of the APIM performed to assess whether positive and negative DC are associated with RS are summarized in Table 4. Regarding the actor–actor effects, patient-perceived NDC is negatively associated with patient RS. Regarding the actor–partner effects, patient-perceived PDC is positively associated with partner RS. Regarding the partner–actor effects, partner-perceived PDC is positively associated with patient RS. No other association is significant.

### 5.3. Moderating Effects of Patients’ and Partners’ Anxiety and Depression on the Relationship between DC and RS

The main results of the APIMoM performed to assess the moderating effects of patients’ and partners’ anxiety and depression on the relationship between positive and negative DC and RS are represented in Table 5. The interaction between patient-perceived positive DC and partner depression is significantly associated with partner RS. The simple slope analysis shows that the effect of patient-perceived positive DC on RS is higher at higher levels of partner depression, although the relationship is significant also at lower levels of partner depression. The interaction between partner-perceived positive DC and patient anxiety is significantly associated with patient RS. The simple slope analysis shows that this relationship is significant only at low levels of patient anxiety. The interaction between patient-perceived negative DC and patient anxiety is significantly associated with patient RS. Similarly, the interaction between patient-perceived negative DC and patient depression is associated with patient RS. In both cases, the simple slope analyses show that these relationships are negative and significant at higher levels of patient anxiety or depression. The coefficients of the simple slope analyses are reported in Table 6.

## 6. Discussion

DC is a key aspect in understanding patients’ and partners’ adjustment to cardiac illness [24] given the beneficial impact of positive DC for individuals and couples in non-clinical samples [64] and in couples coping with several diseases (e.g., [8,21,22]). This study is the first to explore the DC among patients with cardiac disease and their spouses and its link with RS. Furthermore, the dyadic research design provided a much broader understanding of the reciprocal influence patients’ and spouses’ coping has on the other. Moreover, the first study considered the moderating effect of psychological distress in terms of anxiety and depression among cardiac patients and their partners. In fact, cardiac disease is a stressful experience for both patient and partner [8,10,45], causing a significant impact both on the individual and on how the partners interact with each other [52,53,54,55], to such an extent that psychological distress could hinder the benefits of positive support [12,50,65]. The stressful situation for both partners is further complicated by the management of chronicity, which is manifested by problematic and sometimes severe symptoms in heart disease, with significant impairment of daily life, maintenance of drug therapy, diet and exercise routines, for example in heart failure (HF; [66,67]), or in advanced HF when a heart transplant [68] or an artificial heart [69,70] is needed.

Descriptively, it is interesting to note that positive DC perceived by the patient and by the partner are high and negative DC is low, and this suggests that these couples tend to positively use relational skills in their couple relationship also during the initial adjustment to the cardiac illness, probably for three reasons: first, because couples involved in this study already show good functioning, maybe for a selection bias; second, these are stable couples who have been together for several years, so this may have influenced the results; third, following the Post-Traumatic Growth Paradigm [71,72], these couples have been able to find some benefits from the diagnosis of disease, which may have allowed these couples to strengthen their couple bond. One of the most important skills to develop as a couple is the ability to face painful experiences together, take care of each other, and maintain the bond [17].

The differences in means between patient and partner results highlight that perceived positive DC is unbalanced between partners, so partners perceived lower positive dyadic coping than patients and this underlines that the cardiac illness causes asymmetric help. In particular, our result means that partners perceived low DC or that the patient‘s provided support was not recognized by the partner. These findings confirmed the importance of exploring dyadic congruence which could show the cycle process of coping and how the support is shared. Indeed, perceived inequity and lack of reciprocity among partners were found to predict lower couple satisfaction [73], lower quality of life [74] and lower mental health [75]. Conversely, the complementarity of dyadic coping efforts can be functional for the couple‘s well-being [18,73], especially in illness situations [12,13,76]. Our result could be also explained by the fact that partners in our sample mostly comprised women and, according to the literature, women are more relation-oriented and more sensitive than men to the relationship aspects [73].

Regarding the association between DC and RS, findings from this study partially supported our hypothesis, as positive patient- and partner-perceived DC are significantly associated with greater patient and partner RS, conversely, negative patient- and partner-perceived DC with lower patient and partner RS, in line with our first hypothesis, and with the literature showing that the more patients and partners communicate stress and provide supportive behaviors to the other, the more the couple relationship is strengthened and the greater the satisfaction [30].

Our results showed that DC had an effect not only on one’s own perception of RS (i.e., actor effects), but also on the perception of the other (i.e., partner effect). This underlines a dynamic in which the more a partner recognizes and appreciates the role played by the other (for example, by recognizing the practical or emotional support received and showing such an appreciation), the more motivated the other will be in his/her role and he/she will feel involved in a satisfying relationship. For RS, therefore, it is not only important that the other concretely reciprocate the support, but that there is an acknowledgment of what has been done. This follows a logic of gift that is anything but utilitarian, which opens up to trust and re-launches the couple bond in a generative perspective, in line with the Relational–Symbolic Model (e.g., [77,78]).

Findings also showed that partners’ distress had a significant moderating role. The first significant result, regarding positive DC on RS, shows different partner effects: patient-perceived positive DC increases partner RS when partner depression is high and partner-perceived positive DC increases patient RS when patient anxiety is low. These results suggest that when the patient acknowledges the supportive actions of the partner, the latter feels gratified and perceives a more satisfying relationship, especially if the partner—in coping against the stressors of the illness—shows depressive symptoms. This means that similar to studies on cancer patients [21,22], cardiac patients and their partners were more satisfied with their relationships the more they recognized each other’s needs and provided support. Even during a distressing experience such as the illness, having a satisfactory intimate relationship was an interdependent experience, mutually reinforced by each partner. Furthermore, regarding moderators, it seems that anxiety could be a risk factor in this context. In fact, for a low level of anxiety, the positive and strong association between DC and RS is confirmed; high levels of anxiety can make partners less able to seek support from the other or less able to recognize the other’s support provision during DC exchanges, thereby reducing the association between DC and satisfaction for highly anxious partners. Research has shown that anxiously attached people reported that they seek less support overall than people classified as securely attached [79,80]. Conversely and surprisingly, a high level of depression reinforces this link. This may be because expressing difficulties and sadness may foster support provision by the partner as well as feelings of trust, intimacy, and satisfaction in the relationship when support is received. This result contradicts previous findings that confirm a negative correlation between depressive symptoms and relationship functioning [81,82]. The interactional depression theory of Coyne [82] states that depressive individuals use communication and behavior patterns that lead to anger and rejection in the long term as well as lower RS in their spouses. To our knowledge, only one study in the literature found similar results [83] and found that depressed partners could receive even more support, tolerance, and attention from their spouses who do not react negatively, and this could explain the positive effect of depression on relationship functioning.

Finally, two actor effects were found for patient-perceived negative DC; in particular, it is associated with lower patient RS when patient anxiety and patient depression are high. These results suggested that when the patient perceived that his/her partner is ambivalent or hostile and there is a lack of emotional involvement in addition to the patient being anxious or depressed, relationship satisfaction may decrease, and this is in line with the literature demonstrating a detrimental effect of a negative DC response on RS [27]. This underlines that patient distress could harm the quality of support provided and consequently patient outcomes, as assumed in the literature [8,12,84].

These findings show inter- and intrapersonal relations for both patients and partners. Positive DC is beneficial for both patient and partner RS, but these associations vary as a function of their psychological distress. In contrast, perceived negative DC is only significantly associated with lower patients’ RS. Furthermore, we could assume that, on the one hand, partners could be not entirely aware of their negative DC due to the burden of the disease and their new responsibilities; on the other hand, patients could be more vulnerable to negative partner response because they could expect emotional closeness from the other partner and these missed expectations could consequently lead to a lower perception of RS. Furthermore, these expectations could derive from a gender bias according to the literature [85]; in fact, in our study, patients are mostly men and partners are mostly women.

Regarding psychological distress, our findings indicate that distress interacted with both positive and negative DC to play an additional important role in couples’ coping and satisfaction. This highlighted the importance of assessing distress in both patients and partners because providing support for another person in burdensome disease situations is strongly related to personal distress at an individual level.

Understanding how stress impacts close relationships is important because RS is the primary predictor of life satisfaction [34] and it may play a causal role in promoting physical [86], and psychological health [87]. Conversely, marital discord causes deleterious effects on health outcomes [33,88].

From a clinical point of view, this study underlined the importance of examining coping not only as an individual phenomenon pertaining to personal well-being, but within the context of a dyad perspective and as associated with couple’s outcomes [18,89]. Furthermore, showing how both partners can mutually assist each other in the coping process demonstrates how DC in addition to individual coping can be enhanced and fostered by professionals, especially if there is negative DC. In fact, focusing on jointly managing stress, emotional expressions, and communication skills may support a well-functioning relationship. For these reasons, health practitioners working with couples coping with heart diseases are encouraged to utilize empirically validated prevention and intervention programs [39] designed to help couples cope with stress in their relationship during the cardiac disease and to detect the caregivers’ burden, offering specific psychological support to those partners who experience adverse outcomes [90,91].

## 7. Limitations and Strengths

Several limitations of our study should be acknowledged. First, it is a cross-sectional study, and therefore we cannot attest causal effects. Even though the current work was meant to test a specific theoretical model (i.e., Bodenmann’s STM) in the context of cardiac illness, the use of correlational data did not enable to empirically disentangle the role of each variable. For example, it is also possible that partners’ DC moderates the link between partners’ distress and RS. Future studies, using longitudinal or experimental designs, may be of help to this aim. Another limitation is that all instruments are self-report measures that are limited by social desirability and personal biases. Furthermore, participants are predominantly college-educated middle-class people and all Italians, which limits the external validity and generalizability of the findings. The sample is also relatively restricted in terms of age and sex, but these restrictions accurately represent the cardiac population. Moreover, future studies should further investigate this population by exploring the role of a male caregiver, less considered in the literature. Finally, couples involved in this study are mostly long-lasting and functional, so a selection bias should be taken into consideration. Despite these limitations, the current study also has a number of strengths. This study integrated dyadic constructs incorporating both partners’ perspectives, in line with the definition of dyadic research by Kenny [61]. It is the first study on cardiac patients and their partners to consider both individual and relational variables associated with RS. Furthermore, data were collected in different hospitals for generalizing results.

## 8. Conclusions

Our results allowed to describe the patient and the partner as an interactional unit in which positive DC is associated with higher relationship satisfaction. They imply that strengthening positive DC in a couple facing cardiac disease could contribute to them having a well-functioning and sustained relationship. Considering that marital distress is associated with lower recovery paths and poorer outcomes, RS should be considered a relevant goal of interventions in this context. However, interventions specifically aimed at improving RS are relatively scarce [39,92].

Future directions in couple stress research should adopt longitudinal designs to track changes in DC and RS. Furthermore, research should take into consideration all the moderating effects of stress affecting the relationship between partners. Such results are important for clinicians designing couple-based interventions within cardiological care.

## Figures and Tables

**Figure 1 jcm-13-01180-f001:**
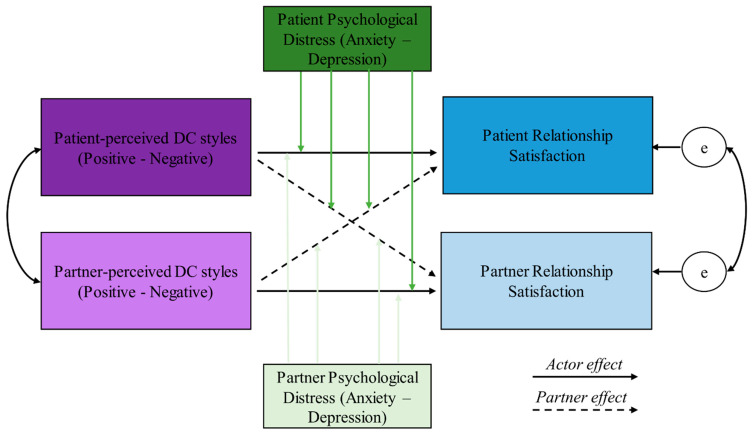
The theoretical model. DC: Dyadic Coping. Green arrows are the moderator effects.

**Table 1 jcm-13-01180-t001:** Characteristics of sample couples (N = 100).

	Patients			Partners		
Variable	M	SD	Range	M	SD	Range
Age	63.41	11.10	34–85	60.50	11.30	31–87
Years of education	14.1	2.9	5–20	14.0	2.9	5–20
Relationship duration	36.34	13.34	3–60	36.34	13.34	3–60
Being parents (%)	88.6			88.6		
First marriage (%)	92.6			89.4		
Male (%)	81.5			18.5		
Employed (%)	51.0			52.3		

**Table 2 jcm-13-01180-t002:** Descriptive statistics.

Measures	Patients M (SD)	Partners M (SD)	Range	Paired-Sample *t*-Test
Perceived Positive Dyadic Coping	3.88 (0.78)	3.36 (0.88)	1–5	*t* (98) = 4.21 ***
Perceived Negative Dyadic Coping	1.90 (0.72)	1.95 (0.69)	1–5	*t* (96) = −0.45
Relationship Satisfaction	6.39 (0.08)	6.23 (0.09)	1–7	*t* (98) = 1.62
Anxiety	1.69 (0.04)	1.80 (0.05)	1–4	*t* (98) = −1.75
Depression	1.67 (0.04)	1.67 (0.04)	1–4	*t* (98) = −0.12

Note. *** = *p* < 0.001.

**Table 3 jcm-13-01180-t003:** Correlations for study variables (*N* = 100 couples).

Measure	1	2	3	4	5	6	7	8	9	10
1. Patient-perceived PDC	-	−0.33 **	0.59 ***	−0.08	−0.06	0.03	−0.04	0.23 *	0.13	0.12
2. Patient-perceived NDC		-	−0.36 **	0.06	0.14	−0.17	0.16	−0.34 **	−0.02	−0.05
3. Patient RS			-	−0.13	−0.03	0.06	−0.04	0.36 ***	−0.05	−0.01
4. Patient anxiety				-	0.70 ***	−0.11	−0.02	−0.18 *	0.09	0.33 ***
5. Patient depression					-	−0.08	0.10	−0.15	0.05	0.274 **
6. Partner-perceived PDC						-	−0.02	0.39 ***	−0.01	0.04
7. Partner-perceived NDC							-	−0.24 *	0.24 *	0.08
8. Partner RS								-	−0.17	−0.116
9. Partner anxiety									-	0.732 ***
10. Partner depression										-

Notes. PDC = positive dyadic coping; NDC = negative dyadic coping; RS = relationship satisfaction. * *p* < 0.05. ** *p* < 0.01. *** *p* < 0.001.

**Table 4 jcm-13-01180-t004:** Summary of the actor and partner effects of the relationship between positive and negative dyadic coping and relationship satisfaction.

	Patient RS		Partner RS	
	B	95%CI	B	95%CI
Model 1				
Patient positive DC	0.044	−0.117–0.205	0.691 ***	0.521–0.861
Partner positive DC	0.393 ***	0.197–0.590	0.314	0.093–0.535
Model 2				
Patient negative DC	−0.458 ***	−0.694–−0.223	0.025	−0.236–0.286
Partner negative DC	−0.268	−0.531–0.005	−0.417	−0.666–0.167

Note. *** = *p* < 0.001. DC = dyadic coping; RS = relationship satisfaction.

**Table 5 jcm-13-01180-t005:** Summary of moderating effects of patients’ and partners’ anxiety and depression on the relationship between DC and RS.

Model	Independent Variable	Moderator Variable	Effect on Relationship Satisfaction			
			Actor Effect		Partner Effect	
			B	95% CI	B	95% CI
Model 1	Patient PPDC	Patient depression	0.035	−0.160–0.231	0.117	−0.122–0.356
	Partner PPDC		0.172	−0.032–0.375	0.226	−0.053–0.505
Model 2	Patient PPDC	Partner depression	0.021	−0.280–0.322	0.315 *	0.027–0.604
	Partner PPDC		−0.210	−0.498–0.079	0.053	−0.243–0.348
Model 3	Patient PNDC	Patient depression	−0.596 **	−1.076–−0.116	−0.148	−0.282–0.067
	Partner PNDC		−0.099	−0.282–0.067	−0.013	−0.388–0.191
Model 4	Patient PNDC	Partner depression	0.056	−0.353–0.465	0.195	−0.220–0.609
	Partner PNDC		−0.438	−0.913–0.036	−0.047	−0.371–0.276
Model 5	Patient PPDC	Patient anxiety	−0.061	−0.326–0.203	0.251	−0.001–0.504
	Partner PPDC		−0.124	−0.345–0.097	0.298 ***	−0.546–−0.050
Model 6	Patient PPDC	Partner anxiety	−0.027	−0.295–0.242	0.049	−0.212–0.309
	Partner PPDC		0.019	−0.207–0.244	0.034	−0.138–0.206
Model 7	Patient PNDC	Patient anxiety	−0.576 ****	−0.931–−0.221	−0.082	−0.361–0.141
	Partner PNDC		−0.131	−0.377–0.116	−0.115	−0.361–0.130
Model 8	Patient PNDC	Partner anxiety	−0.251	−0.702–0.200	0.207	−0.240–0.653
	Partner PNDC		−0.325	−0.689–0.039	−0.227	−0.499–0.104

Note. PPDC: perceived positive dyadic coping; PNDC: perceived negative dyadic coping. * *p* = 0.032; ** *p* = 0.015; *** *p* = 0.018; **** *p* = 0.001.

**Table 6 jcm-13-01180-t006:** Significant moderating effects of anxiety and depression on the relationship between dyadic coping and relationship satisfaction.

	Moderator Values	B	*p*	95% CI
Patient PPDC X HSCL-Dp → RSp	Low	0.429	<0.001	0.187–0.670
	High	0.756	<0.001	0.561–0.950
Partner PPDC X HSCL-AP → RSP	Low	0.368	0.003	0.123 to 0.613
	High	0.034	0.700	−0.138 to 0.206
Patient PNDC X HSCL-AP → RSP	Low	−0.045	0.789	−0.375 to 0.285
	High	−0.692	<0.001	−0.952 to −0.431
Patient PNDC X HSCL-DP → RSP	Low	0.132	0.434	−0.461 to 0.198
	High	−0.749	<0.001	−1.076 to −0.422

Note. The table gives an overview of all significant conditional effects of DC on relationship satisfaction at values of the moderator psychological distress. PPDC = perceived positive dyadic coping; PNDC = perceived negative dyadic coping, RS = relationship satisfaction, HSCL-A = anxiety, HSCL-D = depression, P = patient, and p = partner.

## Data Availability

The data that support the findings of this study are openly available in Zenodo at https://doi.org/10.5281/zenodo.8199693 accessed on 31 July 2023.

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
