# Peer review of "Recognizing and Appreciating the Partner’s Support Protects Relationship Satisfaction during Cardiac Illness"

_jcm, 2024, doi:10.3390/jcm13041180_

Round 1
Reviewer 1 Report
Comments and Suggestions for Authors
The paper focuses on the relation between dyadic coping and relationship satisfaction among couples facing cardiac diseases and shows that positive dyadic coping is associated with a more satisfying relationship with partner and patient psychological distress being a moderating variable affecting this relation. The topic is important and interesting and the paper is relatively well-organized. Therefore I believe it is suitable for publication in Clinical Medicine after major revision as detailed below:
1. The theoretical model in figure 1. should be more precisely explained. Why for example, DC is not the moderator between couples distress and RS? Or the RS is moderating the relation between DC and psychological distress? The causality and the logic of the specific theoretical model that is explored in the paper should be better explained.
2. Why didn’t you include the moderating variable in figure 1? I think it will ease the reading.
3. Line 40 - I suggest replacing the word caregiver to spouse, since the authors refer to couples that live together and a caregiver can be another family member or a payed caregiver.
4. Line 41 - Please explain what you mean in “unbalanced support.
5. Line 50 - The sentence is a bit unclear. Perhaps add the word events after stressors.
6. Lines 55-77 does not include enough references. Is this all taken from Bodenmann (24)? If so, please explain this.
7. Line 140 - Please specify what you mean by different-gender couples. Most couples in the cardiac context are a male patient and a female spouse. If the sample here is different it is important to note it.
8. Line 254-256, please be consistent with using either present or past tense (Patients were men, participants are white).
9. In line 327, the authors mention LVAD, which seems a bit disconnected from the rest of the paper. There are many cardiac illness conditions, why mention LVAD and not the others? I recommend either explaining why LVAD is relevant or review other relevant cardiac conditions.
10. The argument in lines 340-343 is not clear. What is asymmetric help? Maybe you mean that partners perceive the DC as less efficient or beneficial. I believe this finding should be explained and compared with previous literature.
11. Line 331 – why do you refer to initial adjustment and not adjustment in general?
12. Lines 336 – I suggest replace the word benefits, perhaps comfort, relief? I assume no one feels a cardiac disease has benefits.
13. Line 406, I don’t understand the reference here (#65), you refer to your own sample.
14. Line 417 – perhaps replace romantic relationships with dyad perspective. I am not sure romance is the issue here, instead the dyadic lens is important
15. Lines 425-428 – something is wrong with the editing here, the line stopes in the middle and there are different font sizes. The same problem in lines 449-452.
16. Line 440 – What’s a truly dyadic approach? The data collection was individual. Perhaps replace with: the study integrated dyadic constructs, or adopts a dyadic lens.
17. Line 335 – you mention Post- traumatic for the first time in the discussion. Earlier in the paper you referred to anxiety and depression. It is not likely to assume that all couples dealing with cardiac illness has part trauma.
18. Table 1, what does presence of children mean? That the couple have children? If so, than perhaps change to being parents, or having children.
19. I think the discussion lacks a more in-depth explanation with regard to the difference between anxiety and depression as moderators. Why is it that low patient anxiety and high patient depression moderates the relation between DC and RS.
20. Table 1, do the percentage of participants employed include people that already retired or only people that were working at the time of the cardiac event?
21. Here is a list of relevant references, that I believe may be helpful to integrate into the introduction or the discussion:
· Katz, E., Bar-Kalifa, E., Wolf, H., Pietromonaco, P., Hod, H., Klempfner, R., & Vilchinsky, N. (2023). Interpersonal Variables and Caregiving Partners' Burden in Cardiac Illness: A Longitudinal Study. Journal of Social and Personal Relationships, 40(11), 3515-3539.
· Golan, M., Vilchinsky, N. (2023). It takes two to cope with transplantation: The necessity of applying a dyadic approach in the context of artificial heart self-management. Frontiers in Psychology, 14, 1215917.
· Bouchard, K., Greenman, P. S., Pipe, A., Johnson, S. M., & Tulloch, H. (2019). Reducing caregiver distress and cardiovascular risk: A focus on caregiver-patient relationship quality. Canadian Journal of Cardiology, 35(10), 1409-1411.
· Suksatan, W., Tankumpuan, T., & Davidson, P. M. (2022). Heart failure caregiver burden and outcomes: a systematic review. Journal of primary care & community health, 13, 21501319221112584.
· Lum, H. D., Lo, D., Hooker, S., & Bekelman, D. B. (2014). Caregiving in heart failure: relationship quality is associated with caregiver benefit finding and caregiver burden. Heart & lung, 43(4), 306-310.
· Mitra, R., Pujam, S. N. K., Jayachandra, A., & Sharma, P. (2023). Dyadic congruence, dyadic coping, and psychopathology: Implications in dyads for patients with acute coronary syndrome. Journal of Marine Medical Society, 25(Suppl 1), S47-S54.
· Trump, L. J., & Mendenhall, T. J. (2017). Couples coping with cardiovascular disease: A systematic review. Families, Systems, & Health, 35(1), 58.
Comments on the Quality of English LanguageI gave some suggestions on English editing in the comments above.
Author Response
The paper focuses on the relation between dyadic coping and relationship satisfaction among couples facing cardiac diseases and shows that positive dyadic coping is associated with a more satisfying relationship with partner and patient psychological distress being a moderating variable affecting this relation. The topic is important and interesting and the paper is relatively well-organized. Therefore I believe it is suitable for publication in Clinical Medicine after major revision as detailed below:
- The theoretical model in figure 1. should be more precisely explained. Why for example, DC is not the moderator between couples distress and RS? Or the RS is moderating the relation between DC and psychological distress? The causality and the logic of the specific theoretical model that is explored in the paper should be better explained.
We would like to thank the reviewer for this comment. We tried to better explain the goal of our study in the introduction (Lines 95-104 and 109-111). In fact, the main aim is to confirm the strong and positive association between positive DC and RS as well as the strong and negative association between negative DC and RS among the cardiac population using a dyadic lens because this type of data is poorly used especially with chronic diseases. Furthermore, since the literature showed a gap in knowledge regarding the role of psychological distress in moderating the association above, we decided to investigate the interplay among these variables in a sample in which the literature shows both partners suffer from the psychological consequences of the cardiac disease.
- Why didn’t you include the moderating variable in figure 1? I think it will ease the reading.
Thanks, we modify the figure accordingly.
- Line 40 - I suggest replacing the word caregiver to spouse, since the authors refer to couples that live together and a caregiver can be another family member or a payed caregiver.
Thanks for this suggestion. We replace the word caregiver to spouse.
- Line 41 - Please explain what you mean in “unbalanced support.
Thanks for raising this point we replaced the term “unbalanced support” with a periphrases that explains the meaning.
- Line 50 - The sentence is a bit unclear. Perhaps add the word events after stressors.
We would like to thank the reviewer for this suggestion.
- Lines 55-77 does not include enough references. Is this all taken from Bodenmann (24)? If so, please explain this.
Yes, in this section we explain the theory of dyadic coping of Bodenmann. We have considered facilitating the reading by merging the discussion into one paragraph; in the previous version, there were two paragraphs.
- Line 140 - Please specify what you mean by different-gender couples. Most couples in the cardiac context are a male patient and a female spouse. If the sample here is different it is important to note it.
Thank you for the clarification. Yes, the majority of cases of our sample are composed by male patients and female spouses. We replaced different-gender couples with heterosexual couples.
- Line 254-256, please be consistent with using either present or past tense (Patients were men, participants are white).
Thanks for noticing the inconsistency. We have changed the verb tenses accordingly.
- In line 327, the authors mention LVAD, which seems a bit disconnected from the rest of the paper. There are many cardiac illness conditions, why mention LVAD and not the others? I recommend either explaining why LVAD is relevant or review other relevant cardiac conditions.
Thanks for this request, we added specific contributions in lines 335-340 of the discussion.
- The argument in lines 340-343 is not clear. What is asymmetric help? Maybe you mean that partners perceive the DC as less efficient or beneficial. I believe this finding should be explained and compared with previous literature.
We would like to thank the reviewer for this comment. We added a clarification to this point also adding specific literature.
- Line 331 – why do you refer to initial adjustment and not adjustment in general?
We use the term “initial adjustment” for a timing reason because the couples admitted to this study have been coping with heart disease for a relatively short time. In fact, the patients are still in the recovery phase.
- Lines 336 – I suggest replace the word benefits, perhaps comfort, relief? I assume no one feels a cardiac disease has benefits.
This sentence is rephrased accordingly with the comment n.17.
- Line 406, I don’t understand the reference here (#65), you refer to your own sample.
Yes, we referred to our sample based on the literature. For this reason, acknowledging that the sentence might have been unclear as you pointed out to us, we have rephrased it.
- Line 417 – perhaps replace romantic relationships with dyad perspective. I am not sure romance is the issue here, instead the dyadic lens is important
We would like to thank the reviewer for reinforcing the concept we wanted to express.
- Lines 425-428 – something is wrong with the editing here, the line stopes in the middle and there are different font sizes. The same problem in lines 449-452.
Thank you, we corrected it.
- Line 440 – What’s a truly dyadic approach? The data collection was individual. Perhaps replace with: the study integrated dyadic constructs, or adopts a dyadic lens.
We agree and we corrected accordingly.
- Line 335 – you mention Post- traumatic for the first time in the discussion. Earlier in the paper you referred to anxiety and depression. It is not likely to assume that all couples dealing with cardiac illness has part trauma.
Thanks, we agree. For the purposes of the discussion here, it was not deemed necessary to cite the theory that it has been removed.
- Table 1, what does presence of children mean? That the couple have children? If so, than perhaps change to being parents, or having children.
We would like to thank the reviewer for increasing the clarity of the paper.
- I think the discussion lacks a more in-depth explanation with regard to the difference between anxiety and depression as moderators. Why is it that low patient anxiety and high patient depression moderates the relation between DC and RS.
Thank you we added a more in-depth explanation in lines: 356-366; 401-420 and 427-436.
- Table 1, do the percentage of participants employed include people that already retired or only people that were working at the time of the cardiac event?
Thank you for this question. Yes, the participants employed include only people who were working at the time of the cardiac event.
- Here is a list of relevant references, that I believe may be helpful to integrate into the introduction or the discussion:
- Katz, E., Bar-Kalifa, E., Wolf, H., Pietromonaco, P., Hod, H., Klempfner, R., & Vilchinsky, N. (2023). Interpersonal Variables and Caregiving Partners' Burden in Cardiac Illness: A Longitudinal Study. Journal of Social and Personal Relationships, 40(11), 3515-3539.
- Golan, M., Vilchinsky, N. (2023). It takes two to cope with transplantation: The necessity of applying a dyadic approach in the context of artificial heart self-management. Frontiers in Psychology, 14, 1215917.
- Bouchard, K., Greenman, P. S., Pipe, A., Johnson, S. M., & Tulloch, H. (2019). Reducing caregiver distress and cardiovascular risk: A focus on caregiver-patient relationship quality. Canadian Journal of Cardiology, 35(10), 1409-1411.
- Suksatan, W., Tankumpuan, T., & Davidson, P. M. (2022). Heart failure caregiver burden and outcomes: a systematic review. Journal of primary care & community health, 13, 21501319221112584.
- Lum, H. D., Lo, D., Hooker, S., & Bekelman, D. B. (2014). Caregiving in heart failure: relationship quality is associated with caregiver benefit finding and caregiver burden. Heart & lung, 43(4), 306-310.
- Mitra, R., Pujam, S. N. K., Jayachandra, A., & Sharma, P. (2023). Dyadic congruence, dyadic coping, and psychopathology: Implications in dyads for patients with acute coronary syndrome. Journal of Marine Medical Society, 25(Suppl 1), S47-S54.
- Trump, L. J., & Mendenhall, T. J. (2017). Couples coping with cardiovascular disease: A systematic review. Families, Systems, & Health, 35(1), 58.
Thanks for noticing these recent papers added in theintroduction or discussion sections.
Reviewer 2 Report
Comments and Suggestions for Authors
-The topic of the study is very interesting and novel, and is very well presented.
-I would only highlight some small improvements:
*Indicate the software with which the analyzes have been applied.
*Report the results of compliance with the assumptions of the tests applied.
*Rewrite the Conclusions paragraph (lines 389-407) to improve understanding.
*There are typographical errors on lines 426-428 and on lines 448-452.
Author Response
REVIEWER 2
-The topic of the study is very interesting and novel, and is very well presented.
-I would only highlight some small improvements:
*Indicate the software with which the analyzes have been applied.
We would like to thank the reviewer for the precious reading and suggestions. The software is indicated at line n. 263.
*Report the results of compliance with the assumptions of the tests applied.
Thanks, we added the assumptions in a footnote in the method section.
*Rewrite the Conclusions paragraph (lines 389-407) to improve understanding.
Thanks for this suggestion, we rewrite this paragraph at lines 401-420.
*There are typographical errors on lines 426-428 and on lines 448-452.
Thank you, we corrected it.
Reviewer 3 Report
Comments and Suggestions for Authors
I congratulate the authors on a well-planned and relevant study!
With great interest and pleasure I read the article, which was written in compliance with current requirements for scientific journals. The article contains all the necessary sections (Introduction, Methods, Results, Discussion) and the content of the sections fully complies with the regulations.
I believe that the article is almost ready for publication, but there are only 2 comments to improve its quality:
1) In the limitations of the study, the authors rightly point out that “the sample is relatively restricted in terms of age and sex, but these restrictions accurately represented the cardiac population.” However, the authors should pay more attention to the fact that among the patients studied, 81.5% were male. It can be assumed that the results obtained are more consistent with the common case where the cardiac patient is male and his partner is female. However, in couples where the patient is female and the partner is male, the patterns being studied may have a different character. This fact requires more attention and further study. I recommend including this point in the study's limitations. Additionally, I recommend including in the Abstract the percentage distribution of patients and partners by sex
2) Authors should pay more attention to the formatting of the article according to the template, for example, the design of tables and their names, the design of a list of references, etc.
I wish the authors successful revision!
Author Response
We would like to thank the reviewer for the precious reading and suggestions. We added the information about sex percentages in the abstract and we integrated the limitation section accordingly with this appropriate comment. Thank you for raising point N.2, we corrected it.